# Comparison of Experimental Methodologies Based on Bulk-Metagenome and Virus-like Particle Enrichment: Pros and Cons for Representativeness and Reproducibility in the Study of the Fecal Human Virome

**DOI:** 10.3390/microorganisms12010162

**Published:** 2024-01-13

**Authors:** Adriana Soria-Villalba, Nicole Pesantes, Nuria Jiménez-Hernández, Javier Pons, Andrés Moya, Vicente Pérez-Brocal

**Affiliations:** 1Vall d’Hebron Research Institute (VHIR), 08035 Barcelona, Spain; adriana.soria.villalba@gmail.com; 2Area of Genomics and Health, Foundation for the Promotion of Sanitary and Biomedical Research of Valencia Region (FISABIO-Public Health), 46020 Valencia, Spain; nicole.pesantes@fisabio.es (N.P.); nuria.jimenez@fisabio.es (N.J.-H.); javier.pons@fisabio.es (J.P.); andres.moya@uv.es (A.M.); 3Biomedical Research Networking Center for Epidemiology and Public Health (CIBERESP), 28029 Madrid, Spain; 4Institute for Integrative Systems Biology (I2SysBio), University of Valencia and Spanish National Research Council (CSIC), 46980 Valencia, Spain

**Keywords:** virome, RNA, VLP, bulk metagenomics, NetoVIR, SISPA, sequencing

## Abstract

Studies on the human virome based on the application of metagenomic approaches involve overcoming a series of challenges and limitations inherent not only to the biological features of viruses, but also to methodological pitfalls which different approaches have tried to minimize. These approaches fall into two main categories: bulk-metagenomes and virus-like particle (VLP) enrichment. In order to address issues associated with commonly used experimental procedures to assess the degree of reliability, representativeness, and reproducibility, we designed a comparative analysis applied to three experimental protocols, one based on bulk-metagenomes and two based on VLP enrichment. These protocols were applied to stool samples from 10 adult participants, including two replicas per protocol and subject. We evaluated the performances of the three methods, not only through the analysis of the resulting composition, abundance, and diversity of the virome via taxonomical classification and type of molecule (DNA versus RNA, single stranded vs. double stranded), but also according to how the a priori identical replicas differed from each other according to the extraction methods used. Our results highlight the strengths and weaknesses of each approach, offering valuable insights and tailored recommendations for drawing reliable conclusions based on specific research goals.

## 1. Introduction

The human virome, a diverse community of viruses within the human body, remains relatively unexplored compared to the microbiome. Comprising bacteriophages, viruses infecting human cells, and transient viruses, it reflects the diversity of Earth’s viruses. Recent advancements in extraction and detection methods have unveiled this complexity [1,2].

Among the greatest human virome reservoirs, the gastrointestinal tract is the richest due to its diversity and density of viral populations [3]. Interactions between phages, bacteria, and the human host are crucial. Temperate bacteriophages maintain homeostasis, as the phage composition mirrors that of their bacterial prey, but they are also able to disrupt microbial balance. When this happens, a situation of dysbiosis may occur. An imbalance in human microbial communities is being increasingly linked to chronic diseases, infections, and cancer. It has been postulated that some viruses, both resident and transient, can influence tumorigenesis by modifying the microbiome’s predator–prey dynamics [4,5,6,7].

The advent of metagenomic sequencing in the early 1980s marked a significant turning point for microbiomics and viromics, offering a vast field of new possibilities in genomics. Viral populations were first reported in 2002 [8], sparking the exploration of the human virome’s potential. However, metagenomics and NGS analyses provide incomplete viral profiles, and improvements are needed, especially in nucleic acid extraction, to enhance accuracy [9,10].

Viral nucleic acid extraction encompasses unique challenges due to the low abundance of viral genetic material compared to bacterial genomes. Minimizing contaminants and maximizing viral DNA or RNA yield are crucial for accurate quantification and characterization. Furthermore, physical handling of virus-like particles (VLPs) presents another challenge. VLPs and viral particles can be fragile and easily disrupted during extraction, thus potentially leading to the loss of nucleic acids and compromising downstream analyses [11,12]. Furthermore, it should be noted that many viruses are RNA-based; therefore, precautions must be taken to prevent RNA degradation by RNases, necessitating RNase-free procedures [13]. These are two reasons why optimizing extraction and purification methods has become an essential concern in achieving reliable virome sequencing. To address these issues, various nucleic acid extraction methods have been developed, being classified as bulk metagenomes and VLP approaches.

On one hand, bulk extraction captures nucleic acids from the entire microbial community, offering efficiency, easiness, and rapidness, but posing an increased risk of contamination with non-viral genetic material while being computationally demanding [14]. On the other hand, VLP-specific methods are targeted approaches, resulting in purer methods that require less sequencing depth. However, VLP approaches can be biased towards some viruses (due to differences in size, envelope, etc.) and yield lower quantification values, usually requiring amplification steps that may further increase the issue of bias. In addition, they are usually tedious processes. It is still a subject of debate whether choosing one strategy or the other is more optimal for viral nucleic acid extraction [14].

The results of the studies carried out on analyses of the virome are highly dependent on how the analyses were obtained; thus, for any starting sample, factors external to the composition of the sample itself, such as the sample collection, preservation, and handling, or the extraction method used, are critical for a solid analysis. Therefore, establishing the reliability, representativeness, and reproducibility of the data generated is essential for drawing valid conclusions from the obtained results. To accomplish this, we explored the human virome’s complexities, aiming to define the optimal viral nucleic acid extraction protocol that combines purity and reproducibility, ensuring accurate virome analysis.

## 2. Materials and Methods

### 2.1. Study Participants, Sample Collection, Preparation, and Storage

All fecal samples used for this study were previously collected in 2021 as part of the AECC 2017-1485 project awarded to Prof. A. Moya, funded by the *Fundación Científica de la Asociación Española contra el Cáncer.* Aliquots of stool samples remained stored at −70 °C at the facilities of the biobank Biobank IBSP-CV (PT17/0015/0017), integrated in the Spanish National Biobanks Network and in the Valencian Biobanking Network, located at the Foundation for the Promotion of Sanitary and Biomedical Research of Valencia Region (FISABIO-Public Health). Stool samples were selected from previously anonymized samples obtained from a cohort of healthy volunteers from the Valencian Region. They had been characterized by carrying Lynch syndrome mutations, genetically diagnosed at the Program of Genetic Counseling in Cancer of the Valencian Community (Spain). These samples had their corresponding signed informed consent and information sheets. All experimental protocols, as well as the sample cession, were approved by the Ethics Committee for Clinical Research of the Directorate General of Public Health and Center for Advanced Research (CEIC-DSP/CSISP).

Briefly, each participant was provided with a sample collection kit, which included sterile containers containing 2 mL of RNAlater Solution (Ambion, Austin, TX, USA) to preserve and stabilize RNA integrity until arrival at the laboratory. Upon arrival at the laboratory, samples were promptly homogenized by adding 2 mL of phosphate-buffered saline (PBS), consisting of 8 g of NaCl, 0.2 g of KCl, 1.44 g of Na_2_HPO_4_, and 0.24 g of KH_2_PO_4_ per liter (pH 7.2). Subsequently, they were centrifuged at 805× *g* at 4 °C for 5 min to eliminate fecal debris. The 1.5 mL samples of resulting supernatants were then split into 2 mL screw cap microcentrifuge tubes and stored at −70 °C prior to processing.

For the experiments, two tubes containing the frozen fecal solution from each subject were thawed on ice and split into six 500 μL aliquots, spanning two replicas for the three tested extraction protocols. One of them was based on bulk metagenomics (Protocol 1), and two of them were based on a VLP strategy (Protocols 2 and 3).

### 2.2. Nucleic Acid Extraction Procedures

#### 2.2.1. Protocol 1: Bulk Metagenomics Strategy

The first extraction protocol, based on bulk metagenomics, used the QIAamp ^®^Fast DNA Stool Mini (QIAGEN, Valencia CA, USA), following the manufacturer’s instructions with some modifications. In addition to the stool samples, two replicas used as negative controls, containing sterile water instead of stool solution, were run in parallel to the samples throughout the procedure. Briefly, samples were centrifuged at maximum speed (20,000× *g*), the supernatant was discarded, and 1 mL of InhibitEX Buffer and 20 μL of lysozyme were added. The samples were then incubated for 30 min at 37 °C. After the incubation, 200 μL of glass beads, which had been acid-washed (Sigma, St. Louis, MI, USA) and previously prepared, were added to the samples, and the mix was heated for 5 min at 95 °C. The mix was then centrifuged at maximum speed for 1 min. Next, 600 μL of the supernatant was transferred to a 1.5 mL microcentrifuge tube containing 20 μL of proteinase K. After adding 600 μL of Buffer AL, the tubes were incubated at 70 °C for 10 min, followed by the addition of 600 μL of absolute ethanol and mixing by vortex. Afterwards, 600 μL of the sample was transferred to the column and centrifuged for 1 min at maximum speed. This step was repeated until all the samples passed through the column. Then, 500 μL of the first washing buffer, Buffer AW1, was introduced to the column, and 1 min of centrifugation at maximum speed was carried out. This step was repeated with Buffer AW2. Finally, the column was placed into the final collection tube where 50 μL of the elution buffer ATE was added, and, after incubation at room temperature for 1 min, DNA was eluted by centrifugation for 1 min. The fluorometric quantification was then assessed using a Qubit^®^ dsDNA HS (High Sensitivity) Assay Kit (Invitrogen by ThermoFisher Scientific, Waltham, MA, USA) in an Invitrogen™ Qubit™ 3 Fluorometer (Invitrogen by ThermoFisher Scientific), according to the manufacturer’s instructions.

#### 2.2.2. Protocol 2: VLP-Enrichment Strategy A (Modified NetoVIR)

The first of the two VLP strategy protocols tested was based on the Novel Enrichment Technique of Viromes (NetoVIR) [15], a fast, reproducible, and high-throughput technique expressly devoted to NGS gut viromics studies However, it included some modifications affecting the RT-PCR amplification and PCR product purification. As in protocol 1, two negative controls, containing sterile water instead of stool solution, were run in parallel to the samples throughout the procedure, but in this case, two additional negative controls of the RT-PCR reaction were also included by adding sterile water instead of extraction products.

For the enrichment step, the fecal suspension was homogenized and centrifuged at 17,000× *g* for 3 min, retrieving at least 200 μL of the supernatant, which was then filtered in a 0.8 μm filter (Sartorius, Göttingen, Germany) at 17,000× *g* for 1 min. Finally, 7 μL of a premade 20× resolving enzyme buffer (12.11 g of 50 mM Tris, 1.47 g of 5 mM CaCl_2_ and 0.61 g of 1.5 mM MgCl_2_ in 80 mL of ultrapure H_2_O, pH 8.0), 2 μL of benzonase (Sigma-Aldrich, St. Louis, MI, USA), and 1 μL of micrococcal nuclease (Thermo Fisher Scientific) were added to 130 μL of sample filtrate. The mixture was incubated for 2 h at 37 °C. To stop the reaction, 7 μL of 0.2 M EDTA was added. The extraction step of this protocol was optimized using the QIAamp Viral RNA Mini kit (QIAGEN), following the manufacturer’s protocol, but proceeding without the addition of carrier RNA to the lysis buffer at the first step of the extraction. For the amplification by RT-PCR, the components (1 μL of 50 μM random hexamers, 1 μL of 10 mM dNTP mix, 11 μL of template RNA) from the SuperScriptTM IV First-Strand cDNA Synthesis Reaction kit (Invitrogen, Carlsbad, CA, USA) were combined in a reaction tube. This mix was heated at 65 °C for 5 min and incubated on ice for 1 min. Then, 4 μL of 5X SSIV buffer, 1 μL of 100 mM DTT, 1 μL RNaseOUT™ Recombinant RNase Inhibitor, and 1 μL of SuperScriptTM IV Reverse Transcriptase (200 U/μL) were added to the mix and incubated at 23 °C for 10 min, 50–55 °C for 10 min, and 80 °C for 10 min, then finished by heating at 85 °C for 5 min. Afterwards, the second-strand synthesis was achieved using the second-strand cDNA synthesis kit’s (Invitrogen) reagents and following its instructions. First, 20 μL of first-strand cDNA synthesis reaction mixture, 55 μL of nuclease-free water, 20 μL of 5X second-strand reaction mix, and 5 μL of second-strand enzyme mix were pipetted directly into the first-strand reaction tube on ice. Incubation was performed at 16 °C for 60 min, and to stop the reaction, 6 μL 0.5 M EDTA with pH 8.0 was added. The reaction was kept on ice until 10 μL (100 U) RNase was added. Then, it was incubated for 5 min at room temperature. Finally, an additional modification of the original protocol was introduced for the PCR product purification, which was carried out using the DNA Clean and Concentrator-5 kit (Zymo Research, Freiburg, Germany). For each sample, 5 volumes of DNA binding buffer were added, and the mix was transferred to a Zymo-Spin column with a collection tube and centrifuged at 14,000× *g* for 30 s. Subsequently, 200 μL of DNA washing buffer was loaded into the column and centrifuged at 14,000× *g* for 30 s, a step that was repeated once more. Finally, 15 μL of preheated UltraPure DEPC-treated water, heated at 70 °C, was added and an incubation lasting 5 min at room temperature was performed, followed by centrifugation at 14,000× *g* for 1 min. The fluorometric quantification was then assessed using a Qubit^®^ dsDNA HS (High Sensitivity) Assay Kit according to the manufacturer’s instructions.

#### 2.2.3. Protocol 3: VLP-Enrichment Strategy B (Modified SISPA)

The second of the two VLP strategy protocols tested was based on the sequence-independent single-primer amplification (SISPA) technique. It shared the initial steps, that is, the enrichment and nucleic acid extraction, with Protocol 2, as well as the PCR product purification. As in Protocol 2, two negative extraction controls and two additional negative RT-PCR reaction controls were included, but two negative PCR amplification controls were also used for the amplification step of this protocol (see below).

After the enrichment and nucleic acid extraction (see protocol 2 for details), the amplification step was started by mixing 1 μL of random primer A (5′-GTTTCCCAGTCACGATCNNNNNNNNN-3′, Condalab, Torrejón de Ardoz, Spain), 1 μL of dNTPs (Ecogen, Barcelona, Spain), 3 μL Ultrapure DEPC-treated water, and 8 μL of extracted DNA/RNA. This mix was denatured by incubating it for 5 min at 65 °C, then cooled on ice for 5 min. Afterwards, the Reverse Transcriptase SuperScriptTM IV kit (Invitrogen) was utilized according to the manufacturer’s instructions. For each tube, 4 μL of 5× SSIV buffer, 1 μL of 100 mM DTT, 1 μL of RNase™ Out, and 1 μL of SuperScript™ IV enzyme were added. The following reverse transcription conditions were used: 23 °C for 10 min; 50 °C for 10 min; and, finally, 80 °C for 10 min. Subsequently, 1 μL of RNase H (ThermoFisher Scientific) was introduced to the tubes (kept on ice). Then, for the second-strand synthesis, the tubes were incubated at 95 °C for 5 min in the thermocycler and cooled down on ice for 5 min. A Sequenase I mix was prepared with the following reagents: 2 μL of Sequenase reaction buffer, 0.3 μL of Sequenase 2.0 enzyme (ThermoFisher Scientific), and 7.7 μL of UltraPureTM DEPC-treated water. The following conditions were used: from 10 °C to 37 °C for 8 min, ramping up 1 °C every 18 s, and 37 °C for 8 min, then 94 °C for 2 min and 10 °C for 5 min. To complete the second-strand synthesis, the Sequenase II mix was prepared by adding 0.9 μL of enzyme dilution reagent and 0.3 μL of Sequenase 2.0 enzyme in the following conditions: 10 °C to 37 °C for 8 min, ramping up 1 °C every 18 s, then 37 °C for 8 min, 94 °C for 8 min, and 10 °C for 5 min. The amplification step was carried out using samples with a mix of 8 μL of MgCl_2_, 10 μL of PCR Gold Buffer 10X, 1 μL of dNTPs 100 mM, 1 μL of Taq DNA polymerase, 1 μL of Primer B (5′-GTTTCCCAGTCACGATC-3′, Condalab), and 69 μL of UltraPureTM DEPC-treated water. Finally, 10 μL of the sample was added. Tubes were incubated at 95 °C for 10 min, 94 °C for 30 s, 40 °C for 30 s, 50 °C for 30 s, 72 °C for 1 min, and 72 °C for 10 min. Finally, fluorometric quantification was assessed using a Qubit^®^ dsDNA HS (high-sensitivity) Assay Kit according to the manufacturer’s instructions.

### 2.3. Library Preparation and Sequencing

Before construction of the libraries, an automated electrophoresis process using the Agilent 4150 TapeStation System (Agilent, Santa Clara, CA, USA) with the high-sensitivity D5000 ScreenTape system kit, allowed for the assessment of the quality and length of DNA and RNA samples, in this case analyzing DNA molecules from 100–5000 base pairs (bp). Libraries were generated for both the samples and negative controls. Libraries were prepared using Nextera^®^ XT DNA kit (Illumina, Carlsbad, CA, USA) based on fragmentation and tagmentation of the input DNA. The kits employed were the Nextera XT DNA Library Preparation Kit (Illumina) and the Nextera XT Index Kit (Illumina), following the manufacturer’s guide with the following modifications: 10 μL of TD buffer, 8 μL of DNA at 0.2 ng/μL, and 2 μL of tagmentase. The incubation at 55 °C lasted 2 min 30 s, and the clean-up was performed with 0.8× Ampure beads. Furthermore, once this procedure was finished, a quantification using a Qubit^®^ dsDNA HS (high-sensitivity) Assay Kit was carried out. Then, after examining the DNA concentration, samples with undesirable values were selected for a 10-cycle recovery PCR, being purified with 0.9× Ampure Beads. Sequencing was carried out using the NextSeq^®^ 500 System from Illumina, selecting the conditions for the acquisition of a single-read DNA sequence with a length of 150 base pairs using the NextSeq550 MidOutput kit (150 c).

### 2.4. Bioinformatic Analysis

The raw BCL files were converted to standard fastq files by means of the bcl2fastq program (version 2.20.0.422) of Illumina, and the single-end reads were filtered out and trimmed for quality with the Fastp application (version 0.23.3) [16] in four sequential steps: (i) front and tail bases with quality values lower than 20 were trimmed; (ii) the bases lower than 15 on the right of the mean quality in the front to tail sliding window with size 4 were dropped; (iii) ploy X tails were trimmed; and (iv) reads shorter than 50 bases were discarded. The filtered reads were mapped onto the Homo sapiens genome database (GenBank assembly accession GCA_000001405.29, GRCh38.p14 release [17] genome using Bowtie2 (version 2.5.1)) [18] with a very-sensitive-local preset. Kaiju (version 1.9.2) [19] was used for taxonomic assignment of non-human reads by comparing them to the NCBI nr+euk (10 March 2022) reference database, with a maximum of 5 mismatches allowed and a minimum matching length of 20 amino acids. Using R (version 4.1.1) [20], all the reads belonging to the same taxon and sample were counted, and the results were saved in a table. After taxonomic annotation, only those reads matching viruses were further processed. A strict filtering step of the viral reads identified in the negative controls was carried out, so all viruses that had at least one read at the species level in any negative control of their corresponding protocols were removed from the samples.

### 2.5. Statistical Analysis

The obtained abundance matrix at the family level was subjected to statistical analysis using R statistical software v4.2.3 (March, 2023) [20]. The nucleic acid composition of each virus was determined and plotted by protocol and by sample.

To decrease the possible effects of protocol-specific contamination, all taxa identified in the control samples were filtered out from all samples of the respective protocol, and the new abundance matrix was used for all further analyses. Normalization was carried out using the “Analysis of Compositions of Microbiomes with Bias Correction” (ANCOM-BC) package [21]. To delineate differences in the abundances of virus families of each donor attributed to protocol variations, we utilized the normalized matrix and computed the log2 fold-change. To establish the reproducibility of each protocol, using normalized data, the beta diversity among the replicates of each sample was computed using the Bray–Curtis dissimilarity index through the “Vegan” package [22], and a paired Wilcoxon test was used to compare the different protocols using the wilcox.test function from the basic R package. Moreover, to identify the differences between the number of viral taxa detected using each protocol, alpha diversity comparing protocols were determined using the Shannon index and the “Microbiome” package [23].

To further analyze the obtained data, only those samples that had at least one hundred viral reads were considered. Normalization using ANCOM-BC was performed on the remaining samples, followed by conducting an analysis of variance using the distance matrices (Adonis) test in the “Vegan” R package. The differential abundance of taxa between protocols was determined using a Wilcoxon test. The identified taxa were then clustered both by protocol and by similitude.

## 3. Results

### 3.1. Taxonomic Profiling of Metagenomic Sequencing Data

The virome from the fecal samples of ten volunteers recruited in the Valencian Region was isolated using three different extraction protocols. Two replicas (A and B) were processed for each protocol (Table 1). A total of 60 fecal samples and 12 controls were sequenced. A total of 211,711,790 raw single reads were obtained, with 196,963,872 of them remaining after quality control filtering. Of these, 7,611,525 reads of human origin and 119,490,295 corresponding to ribosomal RNA were filtered out. Of the remaining 69,862,052 reads, we were able to assign taxonomy to 48,217,369 reads, of which 2,233,879 corresponded to viruses (Appendix A). After the removal of those taxa that shared reads with their irrespective negative controls, a total of 65,096 reads remained for further processing at the family level (Appendix A). Although 2,159,004 reads were removed from samples during this filtering step, 97.68% of them corresponded to uncharacterized or poorly characterized viruses that had no taxonomic information above the class level, such as unclassified *Caudoviricetes* (46.76%), or were even assigned only as unclassified viruses (20.60%) or unclassified bacteriophages (11.78%).

The comparison of the composition, defined as the number of different viral species identified in each of the three protocols analyzed, is shown in Figure 1, which also depicts their distribution according to their nucleic acid molecules. Protocols 1 (Bulk metagenomics strategy) and 3 (VLP-enrichment strategy A) allowed the greatest number of different virus species to be identified (992 and 972, respectively) compared to Protocol 2 (VLP-enrichment strategy B, 572). The core virus species, that is, those detected by all protocols, consisted of 226 species, all of them dsDNA, ssDNA, and some unclassified/unknown viruses. On the contrary, viruses uniquely identified by Protocols 1, 2, and 3 numbered 496, 101, and 508, respectively. Finally, the numbers of species shared by pairs of protocols were similar in all comparisons, ranging from 109 species (Protocols 2 and 3) to 138 species (Protocols 1 and 2). Interestingly, all viruses identified in the bulk-metagenome-based Protocol 1, either uniquely or shared with one or both of the other protocols, corresponded to dsDNA (902 species) and ssDNA viruses (27 species), with some unclassified/unknown ones making up the remaining viruses identified by this protocol. This distribution contrasts with that of the two VLP based protocols, 2 and 3, which showed fewer species of dsDNA viruses than Protocol 1 (433 and 741, respectively), but were notably enriched in ssDNA viruses (59 and 121, respectively). They also allowed for the identification of viruses that were not identified by Protocol 1, such as RNA viruses (dsRNA, ssRNA) and retroviruses, as Protocol 1 did not carry out a retrotranscription step.

We further identified the viral distribution for each protocol according to the nucleic acids that make up their genome, taking into account not only their presence/absence, but also their relative abundance (Figure 2). Using the taxonomy of the identified viruses as a reference, it was possible to assign to most of them the corresponding nucleic acids, that is, the double-stranded/single-stranded DNA/RNA that constitute their genomes. Thus, the relative abundance of the different nucleic acids could be calculated for each protocol. As expected, with Protocol 1, we were only able to identify DNA viruses, 93.3% of which belonged to dsDNA viral agents. The remaining percentage was distributed between the single-stranded DNA viruses (1.3%) and a remaining unclassified fraction (5.4%). In contrast, Protocols 2 and 3 allowed us to identify both DNA and RNA viruses. Regarding Protocol 2, the percentage of dsDNA viruses decreased to 47%. Compared to Protocol 1, the increase observed in the composition of the ssDNA fraction was corroborated in terms of abundance, reaching up to 4% of the viral reads of this protocol. As for RNA viruses, most of them belonged to the dsRNA group (26%), with single-stranded RNA viruses standing for 1% of the viral reads identified within this protocol. Finally, there was a noticeable shift in the observed pattern when examining the results of Protocol 3. Specifically, its dsDNA fraction (47.2%) and ssDNA fraction (2.4%) resembled those of the other targeted approach. However, dsRNA viruses were barely identified, occupying 0.4% of the reads. On the other hand, the ssRNA fraction was the largest among the three protocols, 15.9%. Comparing the two VLP protocols, Protocol 2 appeared to be better at extracting dsRNA, while Protocol 3 was superior in terms of recognizing ssRNA viruses. Nonetheless, the numbers of unclassified and poorly classified viruses were augmented in both, up to 22% and 34.1% for Protocols 2 and 3, respectively.

The aforementioned results were reinforced by the data shown in Figure 3, where the intestinal viral relative abundance is displayed in terms of virus families and nucleic acid type for each subject, protocol, and replica. As expected, taxonomic differences were observed between individuals for a given protocol, even if the differences were less evident at family level than they would be at the species or even the genus level, as they can be attenuated by the fact that many different virus species among subjects can belong to the same family. In addition, here, we already see differences between protocols within the same individual, as we already noticed for the global outcome (Figure 2) referred to by type of nucleic acid, although here we also provide the taxonomy at the family level. Interestingly, Figure 3 also shows a comparison between replicas of the same protocol. We see that they do not seem to differ excessively, neither in the type of nucleic acid nor the family in the case of protocols 1 and 2, but in protocol 3, a greater variability can be seen between replicates. This qualitative variability was further quantitatively measured (see below). As can be observed, the dsDNA viruses were also the vast majority, although representative variations could be found between individuals. Regarding the diversity of families, the most abundant viruses were those within the groups of the *Caudoviricetes* (dsDNA), *Coronaviridae* (dsRNA), *Alphaflexiviridae* (ssRNA), *Virgaviridae* (ssRNA), and *Microviridae* (ssDNA). Differences in the taxonomic composition at the family level of each volunteer were observed based on the employed extraction protocol (Appendix A).

### 3.2. Replicability Analysis

After normalization using the AMCOM-BC, the difference in the viral composition and abundance between the two replicates of each sample and protocol was calculated using the Bray–Curtis dissimilarity index and compared between pairs of protocols by means of a paired Wilcoxon test (Figure 4). No significant differences were found between Protocols 1 and 2 (*p =* 0.23, Figure 4a), despite being bulk metagenome-based and VLP enrichment-based protocols, respectively. This means that, for some samples, differences between replicas were more pronounced in one protocol compared to the other, whereas for other samples, the opposite was observed, with no clear pattern. However, the other VLP enrichment-based method, Protocol 3, was significantly more variable (less replicable) than both Protocol 1 (Figure 4b) and Protocol 2 (Figure 4c) (*p =* 0.0039 and 0.0059, respectively). This implies that results obtained using Protocol 3 were more replica-dependent than those resulting from any of the other two protocols, something that must be taken into account when designing the experiment.

### 3.3. Diversity Analysis

Furthermore, the alpha diversity of the samples was calculated for each protocol through the Shannon index (Figure 5) and compared between protocols. Protocol 1 showed a significantly lower diversity of families than Protocol 2 (Figure 5a) and Protocol 3 (Figure 5b), as it only extracted DNA viruses, mostly bacteriophages, which fell into a limited number of families. On the other hand, Protocols 2 and 3 showed no significant differences to each other (Figure 5c).

As the number of viral counts was variable among the samples, in order to minimize this effect and to further analyze the obtained data, samples with less than one hundred assigned reads after filtering were removed from downstream analyses. Normalization of the absolute abundance tables through ANCOM-BC was carried out on the remaining samples, and an Adonis test was carried out on the pairs of protocols. Significant differences were found for all comparisons (Protocol 1 vs. Protocol 2: *p =* 0.008; Protocol 1 vs. Protocol 3: *p =* 0.001; and Protocol 2 vs. Protocol 3: *p =* 0.045). Furthermore, a Wilcoxon test was performed to identify the families that were significantly enriched or depleted in each protocol (Figure 6a). Notably, the clustering pattern in the dendrogram shown in Figure 6b illustrates that the sample composition did not group by protocol nor by donor in Protocol 2 and Protocol 3 above all. A higher degree of clustering could be found in Protocol 1, which could be denoting a higher degree of reproducibility. What would, at first, be biologically expected is that the samples would cluster by protocol and by patient. However, it should be taken into consideration that the human virome does not work in entirely the same way as the human metagenome. The interpersonal variability is much greater regarding viral species colonizing human body sites than bacterial ones, a factor that could be interfering with the clustering pattern.

Table 2 presents a comprehensive overview of the three distinct extraction methods employed for virus identification, outlining their respective advantages and limitations. Each protocol’s strengths and limitations lie in its unique approach to virus extraction, ranging from the number of identified taxa and their nucleic acid composition, ease of use, and diversity to replicability, which must be carefully considered in the selection of the most suitable method for virus identification. This summary aims to assist researchers and practitioners in weighing the trade-offs among these extraction techniques to make informed decisions in their virology investigations.

## 4. Discussion

The human virome can be defined as a collection of all viruses settled in a specific niche of the human body, which are part of the human microbiota. Although the bacterial fraction of the microbiota has been more extensively researched than the viral fraction, the significance of the different viruses for human well-being should not be overlooked. Recent advancements in next-generation sequencing (NGS) technologies have enabled the identification of viral sequences within biological samples. However, the study of the human virome still has important drawbacks. In the present study, we compared three different extraction protocols, aiming to identify the pros and cons for their use in virome analysis.

Fecal samples provided by 10 volunteers were extracted using two replicas per sample and three different extraction protocols. The total extracted DNA from a total of 60 samples was sequenced, and the viral data were analyzed.

Among the viral species found in humans, bacteriophages have been identified as the most prevalent viruses in the human digestive tract, while gut phages are predominantly part of dsDNA virus families [8,24,25,26,27]. Furthermore, environmental factors and genetic traits may also contribute to the diversity of phage populations, making viromes highly individual-specific [28]. It is important to determine the abundance of these types of viruses in gut samples. The main weak point of Protocol 1 (bulk metagenomics strategy) is the fact that it does not comprise a reverse transcription step; therefore, it only allows for the sequencing of DNA viruses (mostly dsDNA). This may be considered as a shortcoming for studies aiming at the analysis of the whole viral community, including those other than DNA viruses. However, this methodologically simple protocol, compared to the two VLP-based strategies used in this work, seems to be the most efficient approach in terms of identification of higher numbers of reads attributable to bacteriophage taxa, and, as the majority of bacteriophages are dsDNA viruses, it can be the most recommendable protocol if only bacteriophage identification, although as complete as possible, is sought. In addition, some single-stranded DNA viruses were found in the Protocol 1 outcome, despite the fact that the method for library construction only works on dsDNA. The main explanation for this result is that when the replication of the single DNA strand from these viruses occurs, they incorporate a transient double-strand phase resembling prokaryotic rolling circle plasmids [29]; therefore, those ssDNA viruses were identified at the moment when the nucleic acid extraction was performed. However, the observation of lower levels of ssDNA viruses in Protocol 1 compared to Protocols 2 (VLP-enrichment strategy A) and 3 (VLP-enrichment strategy B) can be explained by the absence of DNase treatment; therefore, no removal of DNA molecules occurred in any protocol. This was combined with the RT-PCR and second strand synthesis steps carried out in Protocols 2 and 3. These additional steps, used to convert ssRNA and dsRNA into ds-cDNA, also resulted in the random priming and amplification of a second strand of ssDNA viruses using their single-strand genomes as templates. Although to a lesser extent, both Protocol 2 and Protocol 3 also showed that dsDNA viruses made up the highest percentage of the identified viruses, making them a reasonable, although less efficient, alternative when phages are searched for along with other types of viruses.

The dsDNA portion was still the most representative viral type found in Protocol 2, and the fraction assigned as unclassified was increased compared to Protocol 1. Interestingly, with this protocol, dsRNA viruses could also be noticed. The contribution of RNA viruses can range from 38 to 63% [30]. Nonetheless, it should be highlighted that the percentage and contribution of the RNA viruses to the total viral population is highly dependent on the environment [30,31]. It is evident that our understanding of RNA viruses is far from complete. Broadly speaking, the gut’s RNA virome has been explored to a lesser extent than the DNA virome. This is primarily attributed to the lack of stability RNA viruses seem to have in samples when compared to DNA viruses, making their identification through metagenomic sequencing challenging [32].

Protocol 3 was the protocol with the greatest number of viruses without complete known classifications. Notably, the identification of dsRNA viruses through Protocol 3 was considerably diminished when compared to Protocol 2. Previous studies have shown that this type of extraction protocol should be efficient in the detection of RNA viruses [33,34]. In 2022, Chrzastek and collaborators were able to complete the whole-genome assembly of SARS-CoV-2 and influenza A from a sample containing a mixture of viruses, and the same was accomplished when applied to avian RNA viruses [34].

The intestinal virome taxa were studied at the phylum and family level. The most abundant intestinal viruses found in this study were bacteriophage members of the group *Caudoviricetes* (dsDNA), regardless of the protocol utilized. These dsDNA viruses are collectively known as “tailed bacteriophages” [35], and could, under specific conditions, alter the intestinal bacterial population, diminishing beneficial bacteria and initiating intestinal inflammation [36]. Furthermore, *Caudoviricetes* represents the majority of phage sequences described to date [27].

Furthermore, other viral families such as *Picornaviridae* (ssRNA), *Alphaflexiviridae* (ssRNA), *Virgaviridae* (ssRNA), and *Microviridae* (ssDNA) have strong representation in the human gut virome, according to the Gut Virome Database (GVD) built by Gregory and colleges [14]. In this study, the database was created from 2697 human gut metagenomes derived from 1986 samples from individuals encompassing 16 countries and extracted by VLP or bulk methods. According to the results, 97.7% of the GVD corresponded to bacterial viruses, and only 2.1% to eukaryotic viruses. In this last group, *Picornaviridae*, *Alphaflexiviridae*, and *Virgaviridae* could be found at the top of the list of the most common viral species. It should be mentioned that the latter two families correspond to plant viruses, but their presence in the human virome is frequent [37]. *Microviridae* was one of the few single-stranded DNA phages to be identified.

At the phylum level, both *Alphaflexiviridae* and *Virgaviridae* belong to the phylum *Kitrinoviricota*, while *Picornaviridae* belongs to the phylum *Pisuviricota*. Both groups include multiple species of viruses that are found in a broad range of environments. A recent phylogenetic study [38] was performed to deeply analyze the RNA virome, and the researchers discovered new clades belonging to both phyla. They also mentioned that *Kitrinoviricota* and *Pisuviricota* showed distinctive genetic and phylogenetic features, so it was possible for them to be related evolutionary groups that are kept apart from other RNA bacteriophages and other RNA viruses. Lastly, the vast majority of single-stranded DNA viruses corresponded to the family *Microviridae* and the phylum *Phixviricota*, icosahedral bacteriophages compromising ssDNA viral species that are utilized as model systems for studying morphogenesis and the evolution of assembly [39].

To determine the reproducibility of each protocol, beta diversity results were compared between replicates. No significant differences were found when comparing Protocol 1 and Protocol 2. However, in comparison to Protocol 3, both Protocol 1 and Protocol 2 exhibited considerably higher reproducibility. Protocol 1, distinguished by its simpler procedure without a PCR amplification step, subjected the samples to minimal manipulation. Protocol 2 was a virus-specific extraction protocol that included a retro-transcription step; nonetheless, the handling of the sample was kept to a minimum. Finally, Protocol 3 contained several amplification steps, increasing the probability of sample contamination. Notably, the amplification primer used in Protocol 3 was a random primer, which might have been responsible for the increase in uncharacterized viruses found using this protocol. Despite the relatively wide use of SISPA-based methods for the characterization of the human virome [40,41,42] to address limitations inherent to virome determination, such as the lack of universal genes equivalent to the prokaryotic 16S rRNA gene, our results on the low replicability of the methodology based on this protocol urge us to be cautious regarding the reliability of this methodology. Theoretically, SISPA-based protocols, such as Protocol 3 in our study, because of their inconsistent reproducibility, require several replicates to be carried out in parallel and pooled. This approach aims to address the significant variability observed among replicates, ultimately aiming for a more precise representation of the actual virome. But, even so, that would still not necessarily reflect the actual composition of the virome due to the randomness of the amplifications in each replica. Even if we were to pool several replicas, the proportions of the different viruses are already distorted and biased. The main advantage of pooling them is that, with the mixture, we can increase the range of viruses detected, but how much exists of each of them would not be quantifiable.

Finally, the heatmap revealed distinct patterns of families’ enrichment or depletion among protocols. For instance, in Protocol 1, the most enhanced groups were *Caudoviricetes* and *Ackermannviridae*, while others, such as *Anelloviridae*, *Picobirnaviridae*, and *Virgaviridae,* were almost depleted. Regarding Protocol 2, *Picobirnaviridae* and *Steigviridae* were the most enriched families, at the expense of *Ackermannviridae* and *Herelleviridae*. Finally, the enriched families in Protocol 3 included those stated in Protocol 2, as well as the families *Herelleviridae* and *Virgaviridae*. It seemed that the patterns for depleted and enriched families between Protocol 1 and Protocol 2 were opposite, while Protocol 2 and Protocol 3 presented similar outcomes. Indeed, studying the maintenance of data between individuals and replicas, both Protocol 2 and Protocol 3 were more unstable than Protocol 1, where preserved results between individuals and duplicates could be observed. These findings were also shown in our sample clustering analysis, where replicas in Protocol 1 were clustered together, whereas Protocol 2 and Protocol 3 duplicates were more widely spread.

## 5. Conclusions

With this work, it has been demonstrated that the election of a certain viral nucleic acid extraction protocol vastly influences the final results. Upon assessing the effectiveness, reproducibility, advantages, and drawbacks of three distinct experimental methods for extracting viral nucleic acids, including two methods specifically targeting VLPs and a third method involving bulk metagenome analysis, it can be concluded that a non-targeted approach, namely, Protocol 1, is advisable in situations where virus diversity is not a significant concern and the project primarily centers on phages. Nevertheless, if the research requires RNA viruses, a virus-specific protocol must be chosen. From the two targeted strategies, Protocol 2 and Protocol 3, it has been demonstrated that the first is the most robust; it showed greater consistency during the reproducibility analysis and correlated better with previous findings, although Protocol 3 successfully identified a greater variety and number of species. Further research in this field is essential in order to assess and choose various extraction procedures, ultimately establishing a specific experimental workflow tailored to the type of virus targeted for detection. In such a manner, outcomes from different projects could be more comparable, and global conclusions could be drawn.

## Figures and Tables

**Figure 1 microorganisms-12-00162-f001:**
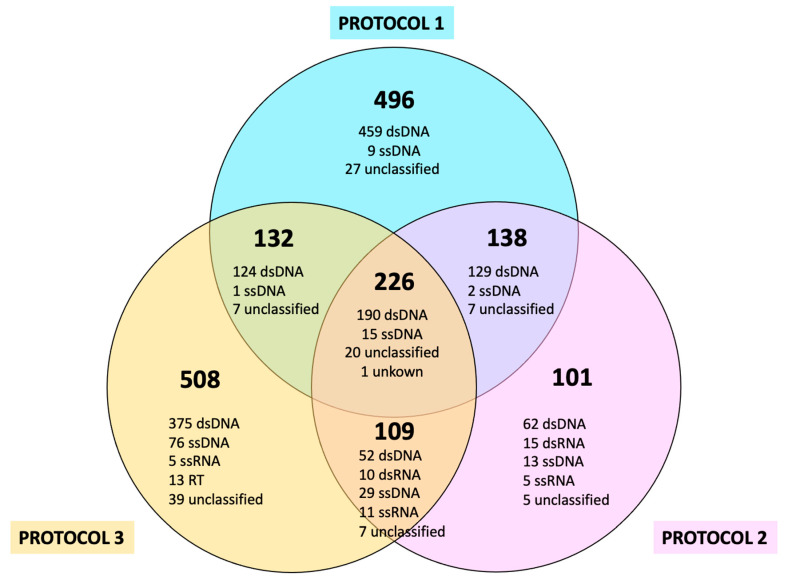
Number of species identified with each protocol. Each circle of the Venn diagram indicates a strategy, while the intersections between circles correspond to the number of species shared among those two protocols, or, in the case of the central intersection, the “core” of species detected through three procedures. In addition, each figure is itemized according to the type of nucleic acid molecule: DNA vs. RNA, single-stranded vs. double-stranded.

**Figure 2 microorganisms-12-00162-f002:**
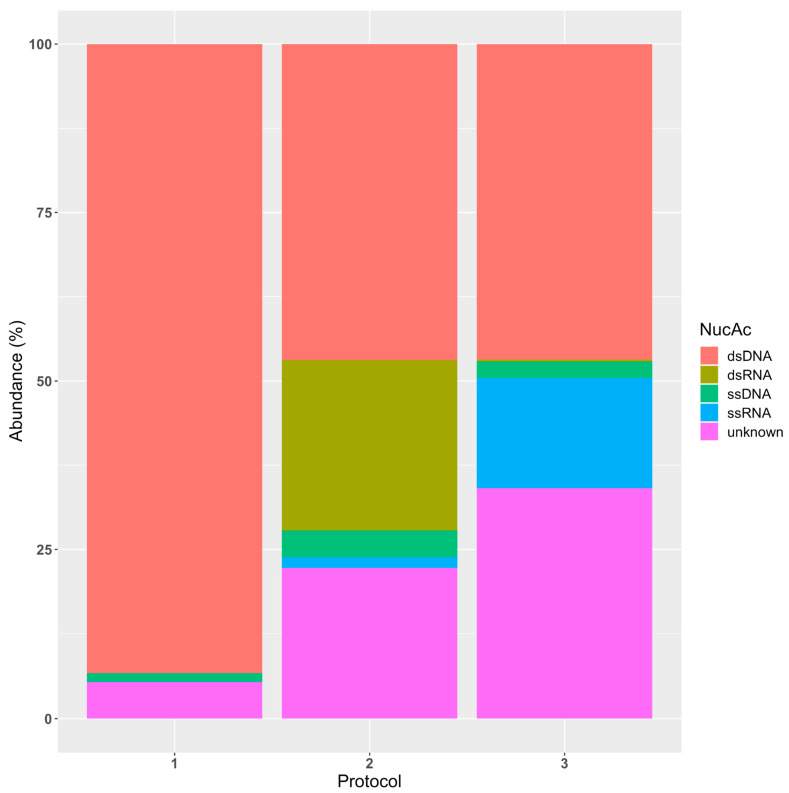
Relative abundances of the different viruses, sorted by the type of viral nucleic acid and found according to the protocol used. The bar graph shows the percentage of double-stranded DNA, single-stranded DNA, double-stranded RNA, and single-stranded RNA detected using Protocols 1, 2, and 3. The unknown fraction includes viral species which could not be taxonomically assigned; therefore, no information about their genome could be obtained.

**Figure 3 microorganisms-12-00162-f003:**
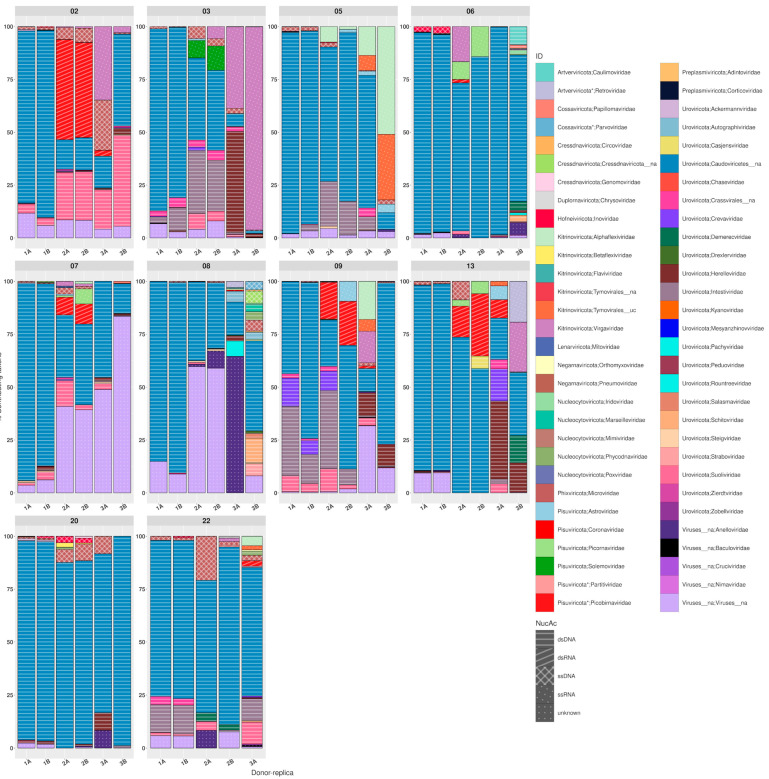
Dominant viral families, including their genomic nucleic acid type, for each subject, identified by protocol and replicas. Data are grouped not only by individual, but also by protocol (1, 2, and 3) and replicas (A or B). Families are marked with colors, while the types of nucleic acid are marked with specific patterns. Details on the relative abundance of viral families can be observed in Appendix A.

**Figure 4 microorganisms-12-00162-f004:**
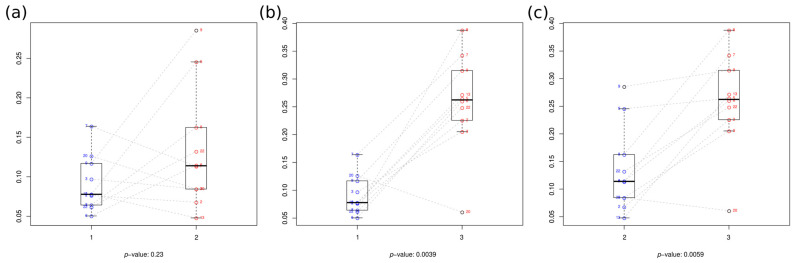
Intra- and inter-protocol consistency by pairs. Boxplots take into account samples, shown as dots, once the mean of the replicas per protocol has been calculated. Comparisons between (**a**) Protocol 1 and Protocol 2 (*p =* 0.23); (**b**) Protocol 1 and Protocol 3 (*p =* 0.0039); and (**c**) Protocol 2 and Protocol 3 (*p =* 0.0059) are shown on the x-axis. The y-axis indicates the Bray–Curtis dissimilarity index. Dashed lines connect the mean values of dissimilarity between each pair of replicas from the same individual in the pairs of protocols under comparison.

**Figure 5 microorganisms-12-00162-f005:**
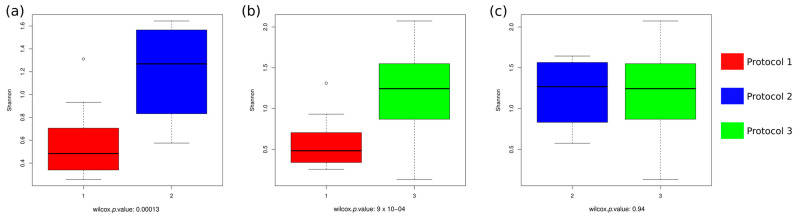
Comparison of the alpha diversity metrics of each pair of protocols. Shannon diversity index was used to estimate the alpha diversity between (**a**) Protocols 1 and 2, (**b**) Protocols 1 and 3, and (**c**) Protocols 2 and 3. Significance of the differences between protocols was estimated by Wilcoxon tests.

**Figure 6 microorganisms-12-00162-f006:**
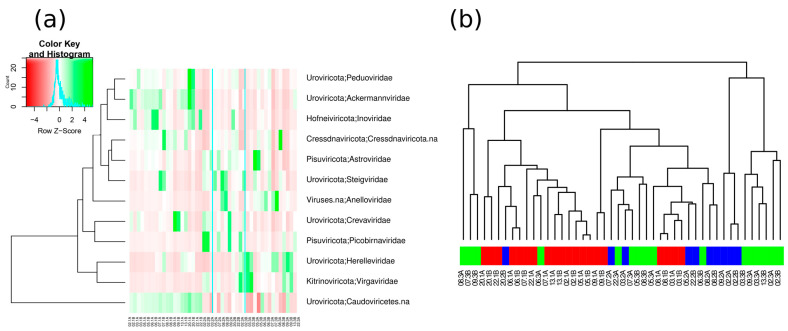
Families enriched or depleted in each protocol. (**a**) Heatmap visualization of the twelve differentially distributed viral families in the samples, sorted by protocol and replica. The heatmap is color-coded to represent the family abundance, with greater enrichment in green and depletion in red. (**b**) Sample clustering by protocol and subject. Red color: Protocol 1, blue color: Protocol 2, green color: Protocol 3. A and B are the two replicas. Only samples with more than viral 100 reads were included in this analysis.

**Table 1 microorganisms-12-00162-t001:** Nomenclature of the anonymized participants in this study, protocols used, and replicas for each sample and protocol used.

Volunteer	Protocol	Replica	ID
02	1	A	06.1A
		B	06.1B
	2	A	06.2A
		B	06.2B
	3	A	06.3A
		B	06.3B
03	1	A	03.1A
		B	03.1B
	2	A	03.2A
		B	03.2B
	3	A	03.3A
		B	03.3B
05	1	A	05.1A
		B	05.1B
	2	A	05.2A
		B	05.2B
	3	A	05.3A
		B	05.3B
06	1	A	06.1A
		B	06.1B
	2	A	06.2A
		B	06.2B
	3	A	06.3A
		B	06.3B
07	1	A	07.1A
		B	07.1B
	2	A	07.2A
		B	07.2B
	3	A	07.3A
		B	07.3B
08	1	A	08.1A
		B	08.1B
	2	A	08.2A
		B	08.2B
	3	A	08.3A
		B	08.3B
09	1	A	09.1A
		B	09.1B
	2	A	09.2A
		B	09.2B
	3	A	09.3A
		B	09.3B
13	1	A	13.1A
		B	13.1B
	2	A	13.2A
		B	13.2B
	3	A	13.3A
		B	13.3B
20	1	A	20.1A
		B	20.1B
	2	A	20.2A
		B	20.2B
	3	A	20.3A
		B	20.3B
22	1	A	22.1A
		B	22.1B
	2	A	22.2A
		B	22.2B
	3	A	22.3A
		B	22.3B

**Table 2 microorganisms-12-00162-t002:** Summary of strong and weak points of the conducted analyses by protocol.

	Number of Identified Taxa	Protocol’s Ease of Use	Nucleic Acid Composition		Replicability	α Diversity
Protocol 1	↑	++	dsDNA	+++	+	−
			ssDNA	+		
			dsRNA	−		
			ssRNA	−		
			unclassified	+		
Protocol 2	↓	+	dsDNA	++	+	+
			ssDNA	+		
			dsRNA	++		
			ssRNA	+		
			unclassified	++		
Protocol 3	↑	−	dsDNA	++	−	+
			ssDNA	+		
			dsRNA	−		
			ssRNA	++		
			unclassified	++		

↑: indicates high number, ↓: indicates low number, and +/− indicates strong and weak points, respectively. When more than one + sign is written (as in ease of use and nucleic acid composition), it indicates the relative intensity of this characteristic compared to other protocols.

## Data Availability

The data for this study have been deposited in the European Nucleotide Archive (ENA) at EMBL-EBI under project accession number PRJEB70723, with sample ID ERS17190394-ERS17190453 and negative control ID ERS17190454-ERS17190465.

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
