# Peer review of "Comparison of Experimental Methodologies Based on Bulk-Metagenome and Virus-like Particle Enrichment: Pros and Cons for Representativeness and Reproducibility in the Study of the Fecal Human Virome"

_microorganisms, 2024, doi:10.3390/microorganisms12010162_

Round 1
Reviewer 1 Report
Comments and Suggestions for Authors
This research is seeking to outline the most effective viral nucleic acid extraction method. The aim of this study is to strike a balance between purity and consistency, ensuring precision in virome analysis, by analyzing the factors beyond the sample's intrinsic composition, such as its collection, preservation, handling, or the technique used for extraction. Hence, ensuring the trustworthiness, inclusiveness, and consistency of the data produced is crucial for deriving credible insights from the findings.
The manuscript is clear, relevant for the field and presented in a well-structured manner.
The manuscript’s results are reproducible based on the details given in the methods section.
The figures and tables are appropriate and easy to interpret and understand.
The data are interpreted appropriately and consistently throughout the manuscript.
The conclusions are consistent with the evidence presented.
More detailed comments are attached

Minor editing of English language required.
Reviewer 2 Report
Comments and Suggestions for Authors
The manuscript by Adriana Soria-Villalba and colleagues reports the results of the comparisons of three bulk-metagenome and virus-like particle enrichment protocols. The authors evaluated the reproducibility of protocols and taxonomic composition of recovered viral groups. The results shown in the manuscript are interesting for researchers working with viruses and metagenomes. Practical recommendations are useful for metagenomic studies. Generally, the narration is consistent and logical, well illustrated, and the paper can be considered for publication. Some notes:
Line 5 - typo (a dot after in the end of title)
Lines 332, 334 - 4(1)% of reads - Does this number correspond to 2,233,879 reads assigned to viruses?
Figure 2 - Please clarify the Y-axis label (per)
Figure 3 - It is an important figure, but the colors are often not distinguishable or hard to match with legends. I think it is necessary to improve the figure. You could show classification on the levels of class and higher in figure 3A and classification of lower-rank taxa on figure 3B. Or you can use taxonomy only on class- and higher ranks. Anyway, this figure is not appropriate and it is very hard to work with it.
Figure 3 - Could you comment on cases of predominance of Inoviridae in some samples?
Figure 6 - I really do not understand the biological meaning of the dendrogram depicted in this figure and conclusions made from the dendrograms. Can you clarify this issue?
Line 524 - typo (a dot after “Protocol 13”)
Line 546 - I would recommend to replace “families” with “groups” (Caudoviricetes is not a family)
